# Efficient Adsorption Capacity of MgFe-Layered Double Hydroxide Loaded on Pomelo Peel Biochar for Cd (II) from Aqueous Solutions: Adsorption Behaviour and Mechanism

**DOI:** 10.3390/molecules28114538

**Published:** 2023-06-03

**Authors:** Yongxiang Huang, Chongmin Liu, Litang Qin, Mingqi Xie, Zejing Xu, Youkuan Yu

**Affiliations:** 1College of Environmental Science and Engineering, Guilin University of Technology, Guilin 541004, China; 19167547675@163.com (Y.H.); qinsar@glut.edu.cn (L.Q.); mingqi109@gmail.com (M.X.); s7777171@outlook.com (Z.X.); 13424351359@163.com (Y.Y.); 2Guangxi Key Laboratory of Theory & Technology for Environmental Pollution Control, Guilin University of Technology, Guilin 541004, China

**Keywords:** Cd (II) adsorption, layered double hydroxide, pomelo peel biochar, mechanism

## Abstract

A novel pomelo peel biochar/MgFe-layered double hydroxide composite (PPBC/MgFe-LDH) was synthesised using a facile coprecipitation approach and applied to remove cadmium ions (Cd (II)). The adsorption isotherm demonstrated that the Cd (II) adsorption by the PPBC/MgFe-LDH composite fit the Langmuir model well, and the adsorption behaviour was a monolayer chemisorption. The maximum adsorption capacity of Cd (II) was determined to be 448.961 (±12.3) mg·g^−1^ from the Langmuir model, which was close to the actual experimental adsorption capacity 448.302 (±1.41) mg·g^−1^. The results also demonstrated that the chemical adsorption controlled the rate of reaction in the Cd (II) adsorption process of PPBC/MgFe-LDH. Piecewise fitting of the intra-particle diffusion model revealed multi-linearity during the adsorption process. Through associative characterization analysis, the adsorption mechanism of Cd (II) of PPBC/MgFe-LDH involved (i) hydroxide formation or carbonate precipitation; (ii) an isomorphic substitution of Fe (III) by Cd (II); (iii) surface complexation of Cd (II) by functional groups (-OH); and (iv) electrostatic attraction. The PPBC/MgFe-LDH composite demonstrated great potential for removing Cd (II) from wastewater, with the advantages of facile synthesis and excellent adsorption capacity.

## 1. Introduction

Heavy metal contaminants in water environments include cuprum (Cu), lead (Pb), zinc (Zn), nickel (Ni), chromium (Cr), cadmium (Cd), mercury (Hg), and arsenic (As) [1]. They are non-biodegradable contaminants that can accumulate in the food chain and cause sickness or contribute to chronic disorders. Cd is one of the most toxic environmental pollutants and is classified as a strong carcinogen by the International Agency for Research on Cancer [2]. The primary sources of Cd pollution are wastewater from the electroplating, mining, smelting, dye, batteries, and chemical industries. Even low concentrations of cadmium-contaminated wastewater may significantly threaten human health through the water–crop–human exposure pathway. The methods for remediating Cd-contaminated water currently include adsorption, membrane separation, chemical precipitation, and the electrochemical method [3,4,5,6]. Adsorption is the most widely used method considering its simplicity, high removal efficiency, and regenerative nature as the adsorbent material. The core of this method lies in the selection of the adsorbents. Several high-efficiency adsorbents are limited in application considering their high prices. For example, activated carbon is expensive and requires chelating agents to enhance the performance of metal sorption, thus increasing the cost of treatment [7]. It has also been found that aerogel has excellent performance in the adsorption of heavy metals [8,9]. Thus, the search for natural, inexpensive, and highly efficient adsorbent materials has now become the focus of research worldwide.

Layered double hydroxide (LDH) is an anionic clay mineral. Allmann tested and confirmed the stratified structure of LDH in 1969 using single-crystal X-ray diffraction. LDHs have several outstanding properties, such as large surface areas, high ion exchange capacities, and strong acid–base buffer lines [10]. Different types of M^2+^/M^3+^-LDH and their modified products have been used to eliminate a wide range of metal (e.g., Cu, Cr, Cd, Pb, and Zn) contamination [11]. Most studies have indicated that the adsorption mechanisms for heavy metals include surface adsorption, ion exchange, complexation, chelation, and coprecipitation of the partially soluble phases [12,13,14]. The heavy metal Cd (II) may be removed by reacting with numerous anions and hydroxyl groups between the LDH layers. It is additionally environmentally friendly as a non-toxic adsorbent material and can be recycled as a renewable adsorbent [15,16,17].

Some experimental studies have demonstrated that the optimal adsorption amount of Cd (II) to LDH-Cl is 61 mg·g^−1^, Cd (II) to MgAl-LDH is 108 mg·g^−1^, and Cd (II) to NiAl-LDH is 10.67 mg·g^−1^ [15,18,19]. However, the locking of Cd (II) adsorption on LDH relies primarily on precipitation and is strongly influenced by the environment and pH of the material’s surface, resulting in poor adsorption stability [20]. The LDH surface is usually positively charged, which reduces the adsorption locking of Cd (II) by electrostatic repulsion, thereby severely limiting the adsorption of heavy metal cations [21]. The adsorption performance may be further improved by functionalising LDH with other materials, meaning their surface physicochemical properties may be modified [21,22]. Biological carbon materials have the advantages of wide sources, excellent pore structures, several active centres, and stable surface charges, and these are ideal materials for the surface functionalisation of LDHs [23]. Simultaneously, in combination with hydrotalcite, biocarbon materials can effectively reduce the agglomeration of hydrotalcite during nucleation and compensate for the poor dispersion, hydraulic conductivity, and electrical conductivity properties of LDHs [24].

However, considering the high cost of their preparation, several highly efficient carbon materials have limited applications. For example, multi- and single-walled car-bon nanotubes (CNTs), carbon quantum dots, graphene, and date palm ash have been studied previously to extend their properties and functionalities [25,26,27,28]. Finding natural, cost-effective, and efficient carbon sources is a key issue in the development of biochar-LDH. The agricultural waste pomelo peel has a large specific surface area and may be transformed into pomelo peel biochar (PPBC) after charring. It is a kind of porous biomass rich in cellulose and pectin [29]. Although the annual output of pomelo peel is huge, only a small amount of pomelo peel is reprocessed into medicine or used as chemical raw material. Most pomelo peel is treated as agricultural waste, which not only causes great wastes of resources but also causes pollution to the environment. According to statistics, the annual output of pomelo in China is more than 4.5 million tons, and pomelo peel accounts for 30–40% of the total quality of pomelo. Thus, the amount of waste pomelo peel produced every year is 1.35~1.8 million tons [30]. This method makes the best use of available resources with minimal impact on the environment. A composite of pomelo peel biochar and LDHs was prepared to produce a pomelo peel biochar/LDH composite material, enabling the efficient adsorption of cadmium-containing wastewater. Many researchers reported biochar and LDH materials for mental removal. Yang Jia [31] synthesised magnetic-biochar-supporting MgFe-LDH composites to remove Pb^2+^ from an aqueous solution. The adsorption process was an aspontaneous endothermic reaction and limited by chemisorption. Tao Wang [32] concentrated on the loading of MnAl-LDH on biochar for Cu (ΙΙ) treatment. The biochar exhibited a higher adsorption capacity than most modified biochars and activated carbons. Compared with these materials, the biochar/LDH shows multiple advantages such as simple syntheses, good adsorption performances, and easy separations. In addition, up to now, the quantitative analysis of heavy metal removal mechanisms has been mainly focused on LDH or BC materials, and there has been little research on other materials, such as BC-LDH composites. Therefore, the combination of biocarbon materials with LDH in a composite material with a novel structure and advanced surface properties is of practical use for engineering purposes.

This study prepared PPBC/MgFe-LDH through co-precipitation to purify Cd (II) from aqueous solutions. These composites were investigated using X-ray diffraction (XRD, X&apos;Pert3 Power, PNAlytical, Almelo, The Netherlands), Fourier transform infrared spectroscopy (FTIR, Frontier, Perkin Elmer, Waltham, MA, USA), Brunauer–Emmett–Teller analysis (BET, ASAP2020M+, Micromeritics corporation, Norcross, GA, USA), a zeta potential analyser (Zeta, Nano ZS 90, Malvern Instruments LTD, Malvern, UK), scanning electron microscopy–energy dispersive spectroscopy (SEM-EDS, Gemini SEM 300, Zeiss, Baden-Württemberg, Germany), and X-ray Photoelectron Spectroscopy (XPS, ESCALAB 250Xi, Thermo Fisher Scientific, Waltham, MA, USA) to study their adsorption performances for Cd (II). The study also explored the influences of the selected materials on the adsorption performances of Cd (II) under different conditions. The relevant adsorption mechanism was additionally analysed and investigated based on the test data. This method made the best use of available resources, with minimal impact on the environment.

## 2. Results

### 2.1. Structure Characterisation

Powder X-ray diffraction characterisation is an important technique for the identification of hydrotalcite-like compounds. Thus, the crystal structures of PPBC, MgFe-LDH, and PPBC/MgFe-LDH were tested (Figure 1a). PPBC displayed a broad and strong diffraction peak around 23°, matching the (002) crystal plane of cellulose, which was typical of amorphous carbon components [31], indicating that there was an evident aromatic carbon structure on the surface [29]. This peak revealed that the pristine biochar possessed the basic characteristics of disordered and turbostratic carbon crystallization [32]. The apparent peaks at 11.061°, 22.294°, 33.972°, 37.900°, and 45.384 °corresponded to the (003), (006), (009), (015), and (018) planes, respectively, indicating the successful synthesis of MgFe-LDH and PPBC/MgFe-LDH. The XRD pattern of MgFe-LDH and PPBC/MgFe-LDH was in good agreement with the standard of Mg_6_Fe_2_CO_3_ (OH)_16_·4H_2_O (TPCDS card No. 00-024-1110) [31], indicating that the combination of MgFe-LDH and PPBC did not destroy the original crystal structure of MgFe-LDH. No apparent PPBC diffraction peak was observed for PPBC/MgFe-LDH, which was primarily attributed to the amorphous structure [33]. In addition, we could calculate the crystallinity and grain size of the prepared nanocomposites. Crystallinity is an important feature affecting a reaction’s performance and removal mechanisms [34]. The crystallinity calculated by software is relative crystallinity. Calculated by jade software (MDI Jade 6, Jade Software Corporation, Christchurch, New Zealand), the crystallinity of PPBC/MgFe-LDH was 28.13%, and the grain size was 124Å. These XRD results revealed that the PPBC/MgFe-LDH composite possessed satisfactory morphology and crystallinity. 

The FT-IR spectra are presented in Figure 1b. PPBC have characteristic peaks of the aromatic groups -CH_2_- (1387 cm^−1^) and C-O (1096 cm^−1^) [29]. MgFe-LDH and PPBC/MgFe-LDH have broad bands at 3436 cm^−1^, which is attributed to the −OH stretching vibrations from the interlayer and the adsorbed water [35]. A bond stretching at 1626 cm^−1^ is due to the bending vibrations of H–O–H (δH–O–H), and it should be assigned to the adsorbed water molecule in the interlayer [36]. Compared with that in PPBC, the existence of interlayer carbonate can be demonstrated via the asymmetric stretching absorption band (ν3) of the CO_3_^2−^ close to 1380 cm^−1^ and 1400 cm^−1^, suggesting the successful modification of the MgFe-LDH on the PPBC [37]. The low-frequency peak around 400–800 cm^−1^ may be related to the lattice vibrations of M–O and O–M–O (M = Mg, Fe) [38]. The FT-IR results confirmed the presence of LDH in ultrafine biochar composites.

We characterised the morphologies and microstructures of PPBC, MgFe-LDH, and PPBC/MgFe-LDH, and the SEM-EDS images are presented in Figure 2. The morphologies and microstructures of the composites indicated that the PPBC (Figure 2a) possessed wavy-like supports on a smooth surface and comprised small biochar particles shaped irregularly [39]. The wavy-like supports were the skeleton of the grapefruit skin biochar that expanded into fractures, and this structure was similar to Wang’s [29] result. According to Figure 2b, the MgFe-LDH was characterised by stacked LDH layers, corresponding with other reported MeFe-LDH adsorbents [14]. The PPBC/MgFe-LDH in Figure 2c showed that the MgFe-LDH was uniformly stacked on the skeleton of PPBC, presenting a fragmented scaly morphology. Figure 2d–i present the EDS micrographs of the PPBC/MgFe-LDH, whose main chemical components were Mg, Fe, C, O, and Cl, which demonstrated that the MgFe-LDH was adequately dispersed on the PPBC. Furthermore, partial surface pores appeared that were filled with nanosized particles. This phenomenon resulted in a significant decrease in the specific surface area of the PPBC/MgFe-LDH composite, consistent with the BET results.

The BET results of the PPBC, MgFe-LDH, and PPBC/MgFe-LDH are presented in Table 1, which were 230.787, 112.716, and 20.995 m^2^·g^−1^, respectively. The pore volume of the PPBC/MgFe-LDH decreased, and the pore size increased to 12.161 nm due to the loading of the MgFe-LDH. The reason for the increased pore size of the MgFe-LDH after loading could be combined with the SEM-EDS results, MgFe-LDH being loaded on the surface and within the pores of the PPBC, thereby reducing the total pore area and pore volume and causing the collapse of the structure after loading when new and larger pores were created. This result may differ from Yang Jia’s synthesis path [31], but the pore sizes of Wang’s [29] grapefruit skin biochar also increased after loading nano-γ-Fe_2_O_3_ particles. The adsorption capacity of the PPBC/MgFe-LDH was notably enhanced with the large arrangement of the MgFe-LDH on the surface of the PPBC.

The elemental compositions of PPBC, MgFe-LDH, and PPBC/MgFe-LDH were analysed, with the results presented in Table 2. For the MgFe-LDH composite with PPBC, the oxygen content increased from 31.440% to 71.470%, and the carbon content increased from 0.740% to 25.210%. This result was similar to Zhou’s study [29]. Thus, it confirmed the successful synthesis of PPBC and MgFe-LDH.

### 2.2. Bath Experiments

#### 2.2.1. Effect of Solution pH

The material’s zeta potential is an important indicator of its stability. As presented in Figure 3a, the pH-Zero point of charge (pHzpc) values of the PPBC, MgFe-LDH, and PPBC/MgFe-LDH are 3.71, 4.11, and 4.03, respectively. The surface of the material is negatively charged when pH > pHzpc. Therefore, in the pH range of 4–7, an increase in pH will promote the adsorption reaction of Cd (II) [40]. The solution pH plays an important role in the distribution of contaminated metal ions and is related to the speciation as well as the charge characteristics of the sorbent–liquid interface [32]. The Cd (II) precipitated when the pH exceeded seven [41]. 

In this study, PPBC, MgFe-LDH, and PPBC/MgFe-LDH were used to adsorb Cd (II) at an initial concentration of 300 mg·L^−1^ from effluents with a solution pH value of 2–7. In terms of Figure 3b, the Cd (II) adsorption capacities of PPBC and MgFe-LDH were close to 40%. Among them, PPBC demonstrated an extremely low capacity at pH = 2, and its removal rate stabilized when the pH increased from 3 to 6. In the range of 4–6, the MgFe-LDH demonstrated a stable removal efficiency. This may have been because electrostatic adsorption was not the main way of adsorption for LDH, and precipitation was the main way. Thus, the change in removal rate was small. PPBC/MgFe-LDH finally exhibited a gradual increase in the pH from 2 to 4 and a sharp increase from 4 to 7. This trend may have been because the adsorbent was densely surrounded by protons at a lower pH, thereby repelling the positively charged Cd (II) and leading to a reduction in binding sites in the PPBC/MgFe-LDH surface. When the pH was greater than 4.03, the PPBC/MgFe-LDH surface carried a negative charge and could effectively adsorb Cd (II) through electrostatic interaction [42]. The diagram of the mechanism is presented in Figure 13(IV). The number of active sites on the PPBC/MgFe-LDH surface increased with the increase in pH and became more positively charged in response to the electrostatic gravitational forces, resulting in the gradual adsorption of Cd (II) [43]. Overall, PPBC/MgFe-LDH exhibited excellent adsorption performance for Cd (II) compared with that of PPBC and MgFe-LDH over a relatively wide pH range.

#### 2.2.2. Effect of Dosage Amount

The dosage of materials is vital for practical applications. The dosage was set at an economical range from 0.5 g·L^−1^ to 3.0 g·L^−1^ to study the effect of the three materials on Cd (II) adsorption. The results are presented in Figure 4. The removal rates of the PPBC, MgFe-LDH, and PPBC/MgFe-LDH increased from 4.36%, 6.46%, and 29.63% to 82.37%, 70.37%, and 95.81%, respectively, at an initial concentration of 300 mg·L^−1^ of Cd (II). Compared with the PPBC and MgFe-LDH, the PPBC/MgFe-LDH had a higher removal efficiency at a lower dose. At 0.5 g·L^−1^, the removal efficiency of the PPBC/MgFe-LDH (29.61111) was four times and five times of that of the PPBC (5.211%) and the MgFe-LDH (7.433%). The adsorption sites of the PPBC/MgFe-LDH were small when the adsorbent dosage was 0.5 g·L^−1^, with a resultant low removal rate. However, the number of adsorption sites increased with an increase in the dosage, along with a rapid increase in the removal rate, which gradually stabilised after the adsorption reached saturation. When the dosage was 1.5 g·L^−1^, the removal rate of the composite material still reached 87%, while the removal rates of the PPBC and MgFe-LDH were only 50.63% and 38.76%. For further investigations on the subsequent kinetics, we selected a dosage rate of 1.5 g·L^−1^ for the experiment. 

#### 2.2.3. Effect of Reaction Temperature

In order to study the influence of the reaction temperature on the adsorption of Cd (II) on the PPBC, MgFe-LDH, and PPBC/MgFe-LDH, simulated wastewater containing Cd (II) (300mg·L^−1^) was prepared, and the adsorption properties of the material on Cd (II) were investigated at 25 °C, 35 °C, and 45 °C, respectively. The results are shown in Figure 5.

As can be seen from the figure, as the temperature increases from 25 °C to 45 °C, the adsorption capacity of the PPBC/MgFe-LDH for Cd (II) increases from 220.5 (±0.329) mg·g^−1^ to 471 (±1.414) mg·g^−1^. The adsorption capacity increases with the increase in temperature, which can be explained by the thermodynamic study in Section 2.2.7. As the ΔG_0_ of the reaction is all negative, its absolute value increases with the increase in temperature; thus, it is more conducive to the reaction.

#### 2.2.4. Sorption Kinetics

Pseudo-first-order and pseudo-second-order equations were used to describe the sorption rate of the solute and determine the optimal adsorption process. As presented in Figure 6, the adsorption kinetics under Cd (II) solution concentrations of 300 mg·L^−1^ indicated that the adsorption capacities of the PPBC and MgFe-LDH increased rapidly over 600 min and gradually levelled off after 1680 min. The PPBC/MgFe-LDH took longer to reach equilibrium, and the final adsorption capacity was about 189.886 mg·g^−1^.

The results are summarised in Table 3. The pseudo-second-order equations of the PPBC, MgFe-LDH, and PPBC/MgFe-LDH better described the adsorption kinetics, as demonstrated by the higher determination coefficient (R^2^) values. Chemisorption was the main form of the adsorption process according to this result. This result was similar to that of Maamoun’s [34]. The PPBC/MgFe-LDH (223.492 (±7.403) mg·g^−1^) demonstrated a higher adsorption capacity than that in the PPBC (90.808 (±0.989) mg·g^−1^) and MgFe-LDH (100.838 (±0.8.971) mg·g^−1^). These results suggested that the overall rate of metal uptake was controlled by the chemisorption process [15]. The superior adsorption performance of PPBC/MgFe-LDH was confirmed by K^2^ that decreased with the material composite, thereby implying that only a few PPBC/MgFe-LDH molecules were required for the elimination of Cd (II) per minute [32].

#### 2.2.5. Intra-Particle Diffusion Model Analysis

The intra-particle diffusion model was the rate-limiting step in the adsorption process. The overall sorption rate was determined by plotting Q_t_ versus t_1/2_. A straight line passing through the origin implied that the rate-limiting stage of the adsorption process was considered as intra-particle diffusion. If it did not pass through the origin, the adsorption process was jointly controlled by the other stages of adsorption [36]. However, considering the difference in mass transfer between the initial and final stages of adsorption, the test results often did not entirely satisfy the ideal situation of fitting a straight line through the origin. 

Moreover, the sorption process was more complicated if the plot exhibited multi-linearity, with each segment reflecting a different mechanism. The first was external surface sorption, often known as instantaneous sorption. The second stage was the gradual sorption stage, in which intra-particle diffusion affected the pace of sorption. The third stage was the final equilibrium stage, in which intra-particle diffusion slowed, owing to the extremely low solute concentration in the solution [44].

As shown in Figure 7, the PPBC and MgFe-LDH adsorption processes may be separated into two stages, namely, instantaneous sorption and gradual sorption. The adsorption process of the PPBC/MgFe-LDH composite was divided into three stages, namely, instantaneous sorption, gradual sorption, and equilibrium. This finding suggested that other rate-regulating steps were present in the adsorption process in addition to particle diffusion. The R^2^ values were high in the early stages (PPBC (0.991), MgFe-LDH (0.984), and PPBC/MgFe-LDH (0.997)), where a sizable mass transfer rate was observed. These values could have been attributed to the Cd (II) concentration of the early-stage solution, which was comparatively high, resulting in a broad concentration gradient, low mass-transfer resistance, and high adsorption rate. As the concentration of Cd (II) decreased, the corresponding concentration gradient and the number of adsorption sites shrank, and the adsorption rate decreased; consequently, the R^2^ decreased (PPBC (0.703) and MgFe-LDH (0.970)). In the third section, the intra-particle diffusion rate of the PPBC/MgFe-LDH became negative (−0.749), indicating that the equilibrium had been attained [15].

#### 2.2.6. Sorption Isotherms

The Langmuir and Freundlich models represent chemisorption and physisorption, respectively. The essence of chemisorption, as opposed to that of physical adsorption, is that the adsorbed molecule forms chemisorption bonds with the solid surface, followed by the exchange, transfer, or sharing of electrons. Chemisorption is, therefore, characterised by a large heat of adsorption, monolayer adsorption, selective and irreversible adsorption, and slower adsorption rates than those of physical adsorption and often requires high temperatures for substantial effects. The results and related parameters are presented in Figure 8 and Table 4.

As shown in Figure 8, higher temperatures lead to higher adsorption capacities of the material for Cd (II). As presented in Table 4, the Langmuir models for the PPBC, LDH, and PPBC/MgFe-LDH fit better than the Freundlich model, indicating that the adsorption processes of the PPBC, MgFe-LDH, and PPBC/MgFe-LDH for Cd (II) were both unimolecular layer adsorptions and primarily chemisorptions. This observation was consistent with the results of the kinetic adsorption. Moreover, the Freundlich parameters 1/n were both less than 1.0, indicating that the adsorption process occurred easily [17]. The maximum adsorbed quantities (Q_m_) for Cd (II) by the PPBC, MgFe-LDH, and PPBC/MgFe-LDH calculated using the Langmuir model were 91.214 (±0.749), 349.732 (±32.665), and 448.96 (±12.385) mg·g^−1^ (45 °C), respectively, with the PPBC/MgFe-LDH adsorbed amounts being greater than those of other LDH-based materials (Table 5). 

#### 2.2.7. Thermodynamic Study

The adsorption isotherms of Cd (II) onto the PPBC, MgFe-LDH, and PPBC/MgFe-LDH were used for the relevant thermodynamic studies at 298, 308, and 318 K, respectively. The calculated values of the ΔG^0^, ΔH^0^, and ΔS^0^ at different temperatures are listed in Table 6. The values of ΔG^0^ became increasingly negative with increasing temperature, indicating that a higher temperature may have facilitated the adsorption of Cd (II) on the composites [45].

As presented in Figure 9, the adsorption behaviour fit well with the thermodynamic equation with a relative coefficient. The adsorption parameters are listed in Table 6. The Gibbs free energies (ΔG) of the three materials were found to be negative, indicating that Cd (II) adsorption was a spontaneous reaction [36]. The absolute value of the ΔG increased with increasing temperature, implying that a relatively high temperature was an advantage for adsorption [46]. In addition, the ΔH^0^ change during the adsorption was positive, indicating that the adsorption of Cd (II) onto PPBC/MgFe-LDH was an endothermic reaction. According to recent reports [47], chemical adsorption dominated the processing when ΔH^0^ ranged from negative to positive. The entropy changes were positive in the present study, suggesting an increasing degree of disorder in the entire system. The amount of free Cd (II) in the system most likely decreased with decreasing concentration after adsorption, resulting in the reduction in electron repulsion in space and an enhancement in the increase in ion chaos. The positive values of ΔS^0^ indicated increased randomness at the solid/solution interface during the adsorption of aqueous Cd (II) on the MgFe-LDH and PPBC/MgFe-LDH [45].

## 3. Adsorption Mechanisms

XRD (Figure 10a), FTIR (Figure 10b), SEM-EDS (Figure 11), and XPS (Figure 12) were used to clarify the interaction mechanisms of Cd (II) with the PPBC, MgFe-LDH, and PPBC/MgFe-LDH and to analyse the resultant surface changes.

Figure 10a demonstrates the XRD patterns of the PPBC, MgFe-LDH, and PPBC/MgFe-LDH in the range of 5−70°. After completing the contaminant capture, the peak intensities of both MgFe-LDH and PPBC/MgFe-LDH were observed to be weakened after adsorption, with minor shifts in the characteristic peaks of the crystalline phase structures of the LDHs. Among them, the diffraction peaks of the MgFe-LDH at 11.001°, 22.092°, 33.415°, 38.289°, and 45.135° corresponding to the (003), (006), (009), (015), and (018) planes, respectively, shifted to higher 2θ angles of 11.341°, 22.783°, 34.142°, 38.542°, and 45.741°. The d_003_, d_006_, d_009_, d_015_, and d_018_ values of the MgFe-LDH decreased from 8.036, 4.020, 2.679, 2.348, and 2.007 nm to 7.796, 3.900, 2.624, 2.334, and 1.982 nm, respectively. The diffraction peaks changed in the same manner as those of the PPBC/MgFe-LDH. The corresponding spacing narrowed, indicating an exchange of the interlayer carbonate ions [45]. Interestingly, new peaks were observed at 17.647° for both the MgFe-LDH and PPBC/MgFe-LDH, which may have been CO_3_^2−^ or OH^−^ between the layers formed by the precipitation of CdCO_3_ or Cd (OH)_2_ [17] (Equations (1) and (2)). The XRD analysis, therefore, demonstrated the importance of precipitation with the interlayer anions (OH^−^, CO_3_^2−^) in the adsorption of Cd (II), as demonstrated in the mechanistic diagram in Figure 13(i).
(1)$${Cd}^{2+}+{OH}^{-}\to Cd(OH{)}_{2}$$
(2)$${Cd}^{2+}+CO_{3}^{2-}\to CdCO_{3}^{2-}$$

The functional groups that participated in the contaminant immobilisation were estimated using the FTIR spectra of the PPBC, MgFe-LDH, and PPBC/MgFe-LDH after adsorption. From Figure 10b, the hydrogen bond stretching peak at 3430 cm^−1^ and 1626 cm^−1^ shifted to 3477 cm^−1^ and 1630 cm^−1^ after Cd (ΙΙ) adsorption, which may have been due to the effect of the hydroxylian, which caused Cd-OH to bind to the LDH (Equation (3)). The shift from 1360 cm^−1^ to 1350 cm^−1^ suggested that CO_3_^2−^ was absorbed by the PPBC/MgFe-LDH, which may have been due to the formation of CdCO_3_ and the anion exchange of OH^-^ with the interlayer CO_3_^2−^ [32]. After the adsorption of Cd (ΙΙ), a new low-frequency peak of carbonates out-of-plane (υ2) was observed at 620 cm^−1^, and the peaks at 630 cm^−1^ were amplified, all of which may have been associated with the formation of Cd(OH)_2_ and Cd_3_(CO_3_)_2_(OH)^2^ [48]. After the adsorption of Cd (ΙΙ), the strength of the absorption band at 780 cm^−1^ in the PPBC/MgFe-LDH spectrum was also observed to be weakened, which may have been caused by surface complexation of Cd (ΙΙ) by functional groups (Fe-O) (Equation (4)). The FTIR analysis, thus, demonstrated the importance of precipitation with the interlayer anions (OH^−^, CO_3_^2−^) and Fe-O functional groups in the adsorption of Cd (II), as demonstrated in the mechanistic diagram in Figure 13(i–iii).
(3)$${Cd}^{2+}+-HO\to HO-Cd$$
(4)$${Cd}^{2+}+F-O\to F-O-Cd$$

SEM-EDS could provide information on various physical and chemical properties of the sample itself, such as morphology, composition, crystal structure, and composition. The microscopic morphologies and elemental compositions of the PPBC, MgFe-LDH, and PPBC/MgFe-LDH samples after Cd (II) adsorption were analysed using SEM-EDS, the results of which are presented in Figure 11. After adsorption, the PPBC appeared to have agglomerates of cadmium precipitates, which may have been formed due to the surface precipitation on the PPBC. Compared with the MgFe-LDH, the PPBC/MgFe-LDH had denser agglomerates after Cd (II) adsorption. In addition, the EDS spectra showed that Mg and Fe were uniformly distributed on the carbon skeleton. The dispersion of Cd (II) was almost identical to that of the Mg-Fe, indicating that the Cd (II) was uniformly loaded onto the MgFe-LDH. PPBC primarily acted as the backbone for MgFe-LDH loading. Therefore, we confirmed that Cd (II) was successfully loaded onto the PPBC/MgFe-LDH. Thus, SEM-EDS again demonstrated the surface precipitation of Cd on the PPBC/MgFe-LDH, as presented in the mechanistic diagram in Figure 13(i). 

The surface compositions and valence states of the PPBC, MgFe-LDH, and PPBC/MgFe-LDH were analysed using XPS (Figure 12a–h). According to Figure 12b, when the Cd 3d peak was deconvoluted, Cd 3d_5/2_ and Cd 3d_3/2_ BEs were at 405.89 and 411.67 eV, respectively, belonging to cadmium hydroxide and carbon hydride, thereby indicating the formation of precipitation in the adsorption process of Cd (II). A similar conclusion was found by Xian Guan [14]. As indicated by the aforementioned results, Cd (II) adsorption occurred via Cd-OH or Cd-O binding to the PPBC/MgFe-LDH, as demonstrated by the XRD pattern. 

The C1s spectrum (Figure 12c,d) and the O1s spectrum (Figure 12e,f) were presented and deconvoluted into three types of C functional groups (C–C/C=C (284.54 eV, 286.64 eV), C–O (286.25 eV, 286.19 eV), and metal carbonate (288.68 eV, 288.68 eV)) and three types of O functional groups (C-O-C (529.76 eV, 529.56 eV), Fe-O-Fe/-OH (531.70 eV, 531.57 eV), and C=O/M-O (533.00 eV, 533.25 eV)), respectively. The C1s XPS spectra after Cd (II) adsorption showed almost no changes in the peaks of the PPBC/MgFe-LDH. However, the rate of metal carbonate increased, which may have been due to the interactions of Cd (II) with the interlayer carbonate ions and surface carbon hydride groups 18. Furthermore, the O1s spectral results indicated that the metal oxide Cd-O increased, and the Fe-O-Fe decreased after adsorption, which may have been due to Fe-O-Fe conversion to Fe O-Cd.

Meanwhile, the Fe 2p_3/2_ and 2p_1/2_ peaks (Figure 12g) were centred at the binding energies of 711.48 and 725.92 eV, respectively, which were typical of Fe (III) in Fe (OH) O. The satellite peaks of the Fe 2p line located at 718.92 further indicated the presence of the Fe (III) species. After Cd (II) adsorption (Figure 12h), a new peak appeared at 711.25 eV in the main Fe 2p_3/2_, which may have indicated the presence of Fe (II). The free FeO- released by the PPBC/MgFe-LDH surface could, therefore, autonomously capture Cd (II), owing to the electrostatic attraction to form FeOCd^+^ ions. The generated cations were, subsequently, precipitated via abundant O-H groups on the material surface to form stable FeOCdOH inner sphere complexation ligands [49]. This process acted as a self-oxidation-reduction reaction (Equation (5)) [47].
(5)$$FeO^{-}+{Cd}^{2+}+{OH}^{-}\to FeOCdOH$$

According to the above XPS pattern, Cd (II) was eliminated from the solution by the isomorphic substitution of Fe by Cd (II), as demonstrated in the mechanistic diagram in Figure 13(ii). Surface complexation with abundant hydroxyl groups on the surface of the material and chemical deposition with free hydroxyl groups (–OH, Fe-O) in solution is presented in Figure 13(iii).

Finally, in combination with the adsorption experiments and a series of characterisations of the material before and after adsorption, we concluded that the adsorption mechanism is adequately presented in Figure 13. The uptake of metal ions is significantly influenced by (i) hydroxide formation or carbonate precipitation, (ii) the isomorphic substitution of Fe by Cd (II), (iii) the surface complexation of Cd (II) by functional groups (–OH, Fe-O), and (iv) electrostatic attraction.

## 4. Materials and Methods

### 4.1. Chemicals

Pomelo peel was obtained from the Guangxi Zhuang Autonomous Region, China. Analytical level magnesium chloride hexahydrate (MgCl_2_·6H_2_O), ferric chloride hexahydrate (FeCl_3_·9H_2_O), sodium hydroxide (NaOH), hydrochloric acid (HCl), and calcium chloride anhydrous (CdCl_2_) were purchased from Yi En Chemical Technology Co., Ltd. (Shanghai, China). 

### 4.2. Preparation of Materials

(1) Preparation of biochar (PPBC) 

The washed pomelo peel washed by deionized water, followed by oven drying at 60 °C for 24 h, and passed through a 60-mesh sieve. The resulting powder sample was placed in a tube furnace with a nitrogen atmosphere at 800 °C, heated at a rate of 5 °C·min^−1^, and cooled naturally after constant temperature for 4 h to obtain the original biochar. The biochar was then stored and preserved in a glass dryer. 

(2) Preparation of MgFe-LDH.

The MgFe-LDH was prepared using a coprecipitation method. Firstly, 0.075 mol of MgCl_2_·6H_2_O and 0.025 mol of FeCl_3_·9H_2_O (n (Mg^2+^): n (Fe^3+^) = 3:1) were placed in a 1 L beaker, to which 500 mL of deionised water was added. To this solution, 1 mol·L^−1^ of NaOH solution was added dropwise until a pH range of 9–10 was attained. The solution was then stirred with a magnetic stirrer for 1 h. The temperature was increased to 80 °C, and the solution was aged for 24 h. Once the reaction was completed, the resulting precipitate was washed by centrifugation several times to remove the excess alkali solution, dried in a blast-drying oven for 24 h, and then ground and sieved. The resultant product procured was MgFe-LDH.

(3) Preparation of PPBC/MgFe LDH

The PPBC/MgFe-LDH composites were prepared using a co-precipitation method. Next, 0.075 mol of MgCl_2_·6H_2_O, 0.025 mol of FeCl_3_·9H_2_O (n (Mg^2+^):n (Fe^3+^) = 3:1), and 5 g of biochar (a solid-to-liquid ratio of 1:100) were placed in a 1 L beaker, to which 500 mL of deionised water was added. Subsequently, 1 moL·L^−1^ of NaOH solution was added to the solution dropwise to a pH range of 9–10. After stirring with a magnetic stirrer for 1 h, the temperature was increased to 80 °C and aged for 24 h. After the reaction was complete, the resulting precipitate was washed by centrifugation several times to remove the excess alkali solution, dried in a blast-drying oven for 24 h, and then ground and sieved. The resulting product procured was the PPBC/MgFe-LDH composite. The material preparation process is shown in Figure 14.

### 4.3. Bath Adsorption Experiments

The experiments were conducted in centrifuge tubes containing 30 mg of PPBC, MgFe-LDH, and PPBC/MgFe-LDH with 30 mL of heavy metal solution. The tube was placed in a thermostatic oscillator and shaken at 220 rpm and 25 °C. The supernatant was then filtered through a 0.45 µm pore-size membrane, and the concentration of Cd (II) was determined using inductively coupled plasma emission spectrometry. The adsorption capacity Q_e_ (mg·g^−1^) and removal efficiency (%) of the adsorbent for Cd (II) was calculated by mass balance, as expressed in Equations (6) and (7).
Q_e_ = V× (C_0_ − C_e_)/m (6)
Removal rate (%) = 100 × (C_0_ − C_e_)/C_0_
(7)
where C_0_ and C_e_ are the initial and equilibrium concentrations (mg·L^−1^) of adsorbate in solution, respectively; V is the volume of solution (mL), and m is the adsorbent mass (mg).

### 4.4. Adsorption Model Fitting

(1) Kinetic model

In order to determine the key step of Cd (II) adsorption process, the adsorption behaviour of different initial Cd (II) concentration was analysed by three kinds of kinetic models, including pseudo-first order model, pseudo-second order model, and Internal diffusion model [24]. They were mathematically represented by Equations (8)–(10), as follows:(8)$$\mathrm{Ln}\left(Q_{e}-Q_{t}\right)=lnQ_{e}-K_{1}\times t$$
(9)$$\frac{t}{Q_{t}}=\frac{1}{K_{2}{Q_{e}}^{2}}+\frac{t}{Q_{e}}$$
(10)$$Q_{t}=K_{d}\times t^{\frac{1}{2}}+C$$

(2) Isotherm model

Adsorption isotherms are essential for understanding the dynamic interaction of adsorption processes. The Langmuir (Equation (11)) and Freundlich models (Equation (12)) and are the most often used adsorption isotherms for analysing the absorption process [15].
(11)$$Q_{e}=\frac{Q_{m}\times K_{L}\times C_{e}}{1+K_{L}\times C_{e}}$$
(12)$$Q_{e}=K_{f}\times {C_{e}}^{\frac{1}{n}}$$
where Q_t_ (mg·g^−1^) is the amount of PPBC/MgFe-LDH adsorbed at time t; Q_e_ (mg·g^−1^) is the equilibrium adsorption amount of PPBC/MgFe-LDH; k_1_ (min^−1^) is the primary rate constant of adsorption; k_2_ (g·(min·mg)^−1^) is the secondary rate constant of adsorption; K_L_ (L·mg^−1^) and K_f_ (mg ^(1−n)^ L^n^·g^−1^) represent the Langmuir bonding term related to interaction energies and the Freundlich affinity coefficient, respectively; Q_m_ (mg·g^−1^) denotes the Langmuir maximum capacity; C_e_ (mg·L^−1^) is the equilibrium solution concentration of Cd (ΙΙ); n is the Freundlich linearity constant; and k_d_ (mg·(min^1/2^·g)^−1^) represents the intra-particle diffusion rate constant.

(3) Thermodynamic study

The thermodynamic parameters, including the standard Gibbs free energy (ΔG^0^), enthalpy (ΔH^0^), and entropy (ΔS^0^), are described by Equations (13)–(15) [50].
(13)$$\Delta G^{0}=-RTlnK_{d}$$
(14)$$\Delta G^{0}=\Delta H^{0}-T\Delta S^{0}$$
(15)$$lnK_{d}=\Delta S^{0}/R-\Delta H^{0}/RT$$
where R is the universal gas constant (8.314 J·mol^−1^·K^−1^), T is the temperature (K), and K_d_ is the equilibrium constant at different temperatures. The K_d_ values are estimated as dimensionless parameters by multiplying the Langmuir constant (k_L_) with the water density. ΔH^0^ and ΔS^0^ were calculated from the slope and intercept of the linear plot of lnK_d_ versus 1/T; ΔH^0^ and ΔS^0^ are the standard enthalpy and standard entropy, respectively.

### 4.5. Characterisation Methods

The PPBC, MgFe-LDH, and PPBC/MgFe-LDH were characterised using X-ray diffraction (XRD) patterns, which were obtained using an X’Pert 3 Powder diffractometer (X’Pert3 Power, PNAlytical, The Netherlands) with copper K_a_ radiation (l ¼ 1.54059 Å). Fourier transform infrared spectroscopy (FT-IR) was used to observe the structural changes in the particle surface (Frontier, Perkin Elmer, USA). The specific surface areas were determined using the Brunauer–Emmett–Teller (BET) method (ASAP2020M+, Micromeritics corporation, USA). A Nano ZS 90 type nanoparticle size and zeta potential analyser was used to analyse and test the zeta potential of the materials (Nano ZS 90, Malvern Instruments LTD, UK). The surface physical morphology and microstructure of the clay composites were determined by a scanning electron microscopy equipped with an energy dispersive spectrometer (EDS) (SEM, Gemini SEM 300, Zeiss, Germany). The surface elements species were analysed using X-ray photoelectron spectroscopy (XPS, ESCALAB 250Xi, Thermo Fisher Scientific, USA).

## 5. Conclusions

In this study, a highly effective PPBC/MgFe-LDH composite adsorbent was successfully prepared via a coprecipitation process to eliminate Cd (II) ions from waste samples using laboratory protocols. MgFe-LDH loading on the PPBC may be considered a good solution for improve MgFe agglomeration and may be dispersed in aqueous solutions, allowing the material to adsorb more effectively. The results of XRD and FTIR showed that the composite was successfully synthesised, PPBC intercalated effectively and increased the strong surface functional groups, SEM could observe the structural characteristics of the composite materials, and BET showed that the specific surface area of the composite was smaller, and the pore size was larger, etc. The optimal conditions for uptake were attained at pH 5.5, with an optimum dose of 1.5 g·L^−1^, temperature 45 °C, and a reaction time of approximately 1660 min. The kinetic parameters fit well with the second-order model, with a correlation coefficient (R^2^) of (0.989) for Cd (II). Moreover, the equilibrium adsorption parameters perfectly followed the Langmuir model, suggesting that the sorption process was related to monolayer adsorption. The highest Langmuir adsorption obtained was 448.960 mg·g^−1^. The series characteristics additionally demonstrated that the adsorption process was spontaneous, endothermic, and unpredictable, indicative of a chemical reaction.

Combined with the results of characterization after adsorption, PPBC/MgFe-LDH could uptake metal cation mechanisms: (i) hydroxide formation or carbonate precipitation, (ii) the isomorphic substitution of Fe by Cd (II), (iii) the surface complexation of Cd (II) by functional groups (–OH, Fe-O), and (iv) electrostatic attraction. Feasibility studies finally demonstrated that the newly developed adsorbent had the potential to be used in the treatment of wastewater for the removal of heavy metals in industrial settings.

## Figures and Tables

**Figure 1 molecules-28-04538-f001:**
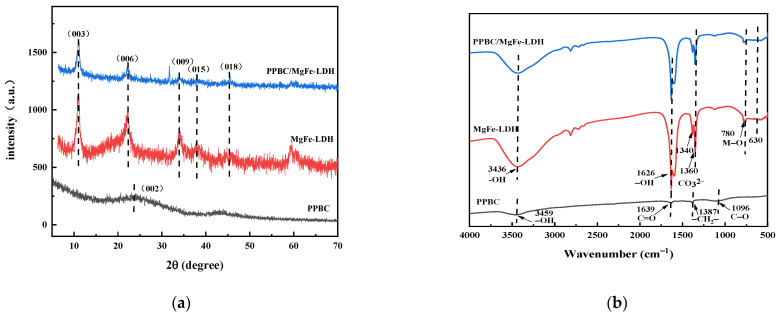
(**a**) XRD spectra of PPBC, MgFe-LDH, and PPBC/MgFe-LDH. (**b**) FT-IR of PPBC, MgFe-LDH, and PPBC/MgFe-LDH.

**Figure 2 molecules-28-04538-f002:**
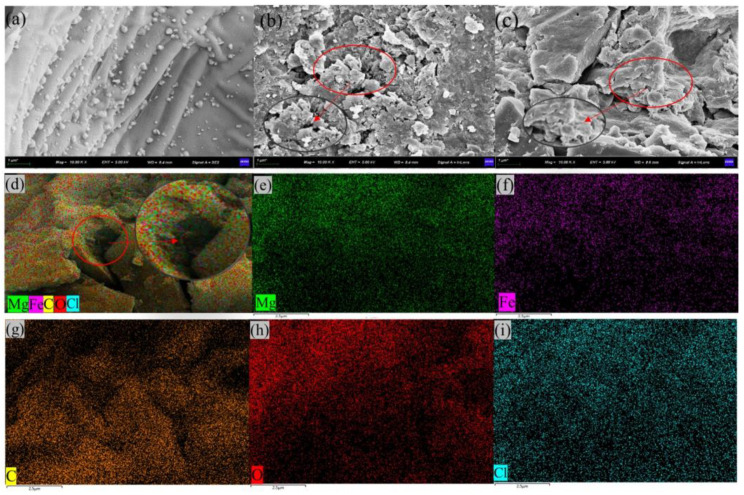
SEM images of (**a**) PPBC; (**b**) MgFe-LDH; (**c**) PPBC/MgFe-LDH; (**d**) EDS images of PPBC/MgFe-LDH; (**e**) Mg element; (**f**) Fe element; (**g**) C element; (**h**) O element; (**i**) Cl element.

**Figure 3 molecules-28-04538-f003:**
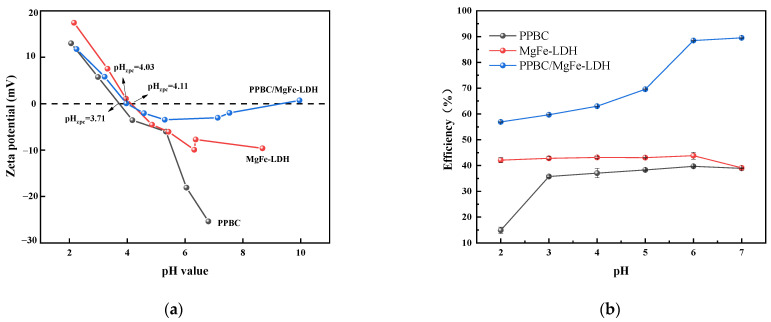
(**a**) pH-zeta of PPBC, MgFe-LDH, and PPBC/MgFe-LDH; (**b**) Effect of initial pH on the adsorption of Cd (II) on PPBC, MgFe-LDH, and PPBC/MgFe-LDH.

**Figure 4 molecules-28-04538-f004:**
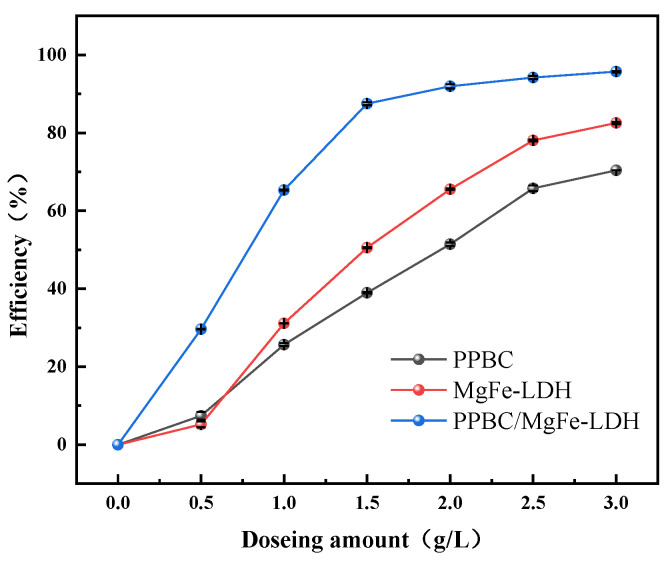
The influence of dosing amount on adsorption of Cd (II) on PPBC, MgFe-LDH, and PPBC/MgFe-LDH.

**Figure 5 molecules-28-04538-f005:**
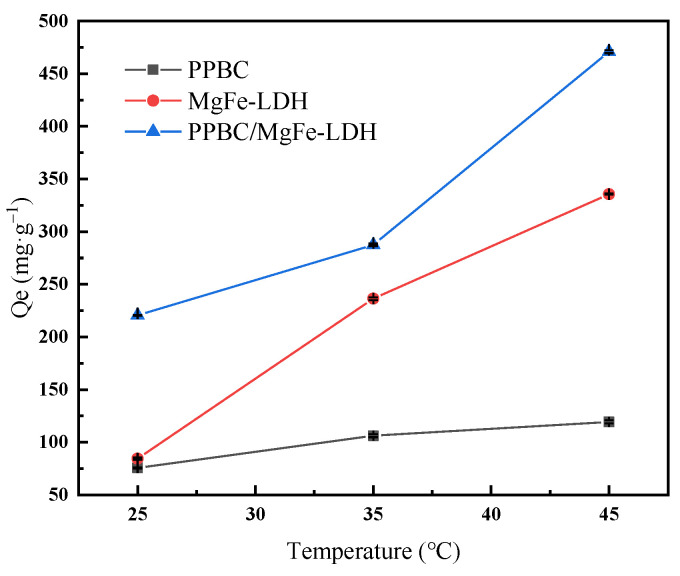
The influence of reaction temperature on adsorption of Cd (II) on PPBC, MgFe-LDH, and PPBC/MgFe-LDH.

**Figure 6 molecules-28-04538-f006:**
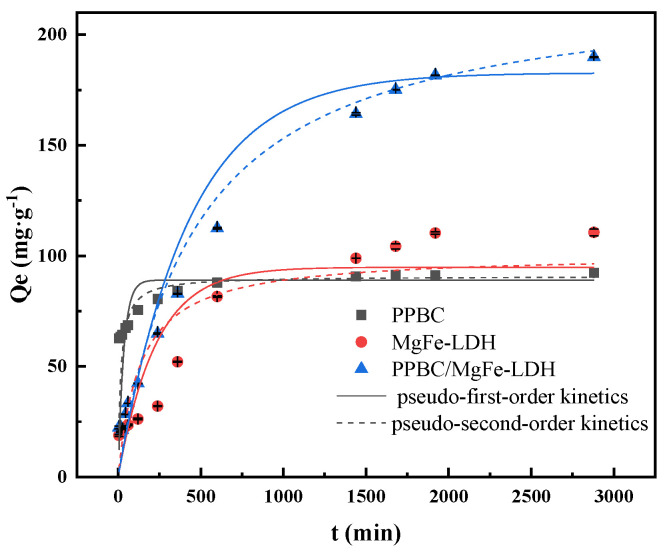
Sorption kinetic fittings of Cd (II) on PPBC, MgFe-LDH, and PPBC/MgFe-LDH. (The dotted and solid lines of different colours correspond to the dynamics of the different materials).

**Figure 7 molecules-28-04538-f007:**
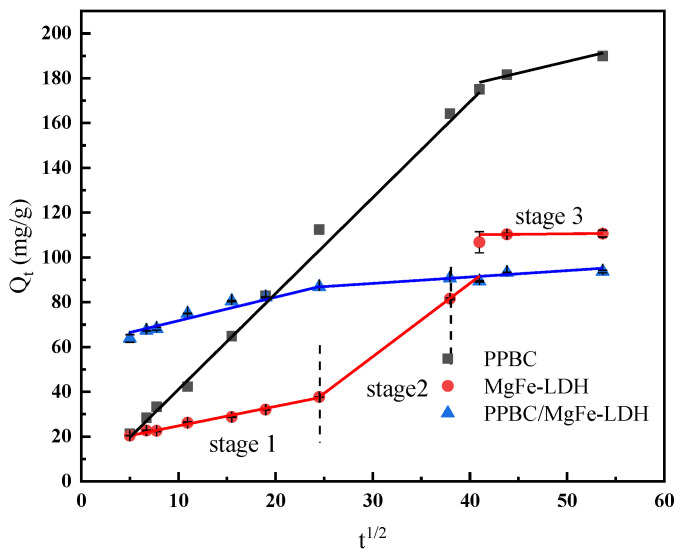
Intraparticle diffusion empirical equations.

**Figure 8 molecules-28-04538-f008:**
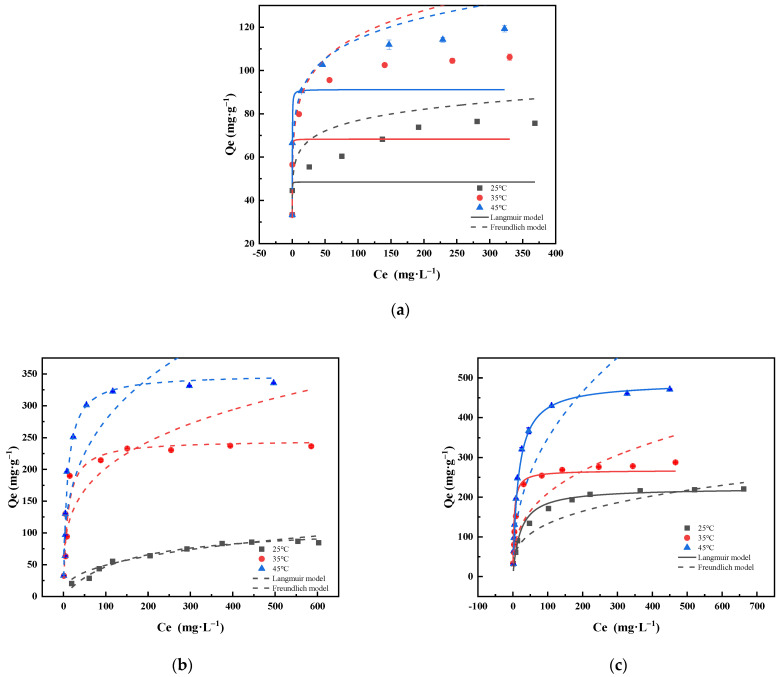
Adsorption isotherms of Cd (II) onto (**a**) PPBC, (**b**) MgFe-LDH, and (**c**) PPBC/MgFe-LDH at 298K, 308 K, and 318 K. (The dotted and solid lines of different colours correspond to the dynamics of different temperatures).

**Figure 9 molecules-28-04538-f009:**
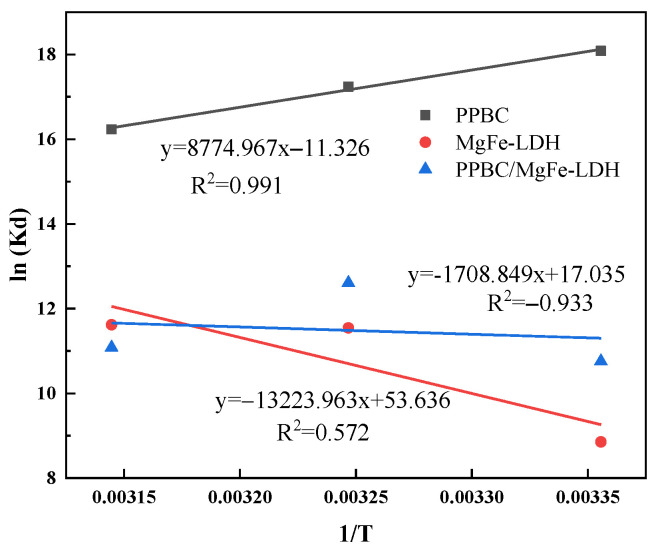
Adsorption thermodynamics of Cd (II) on PPBC, MgFe-LDH, and PPBC/MgFe-LDH.

**Figure 10 molecules-28-04538-f010:**
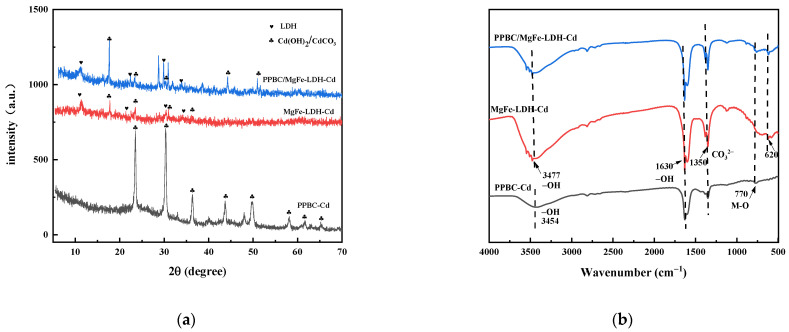
(**a**) XRD pattern of PPBC, MgFe-LDH, and PPBC/MgFe-LDH after Cd (II) adsorption; (**b**) FT-IR of PPBC, MgFe-LDH, and PPBC/MgFe-LDH after Cd (II) adsorption.

**Figure 11 molecules-28-04538-f011:**
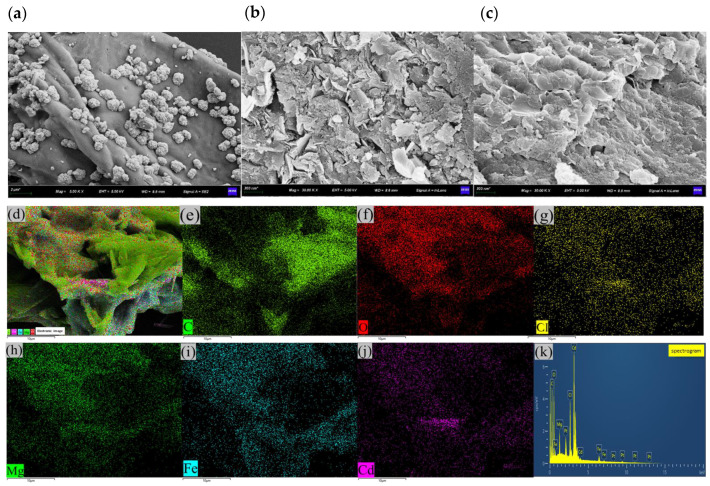
SEM images of (**a**) PPBC, (**b**) MgFe-LDH, and (**c**) PPBC/MgFe-LDH after adsorption; (**d**) EDS images of PPBC/MgFe-LDH after adsorption; (**e**) C element; (**f**) O element; (**g**) Cl element; (**h**) Mg element; (**i**) Fe element; (**j**) Cd element; (**k**) elemental spectrum.

**Figure 12 molecules-28-04538-f012:**
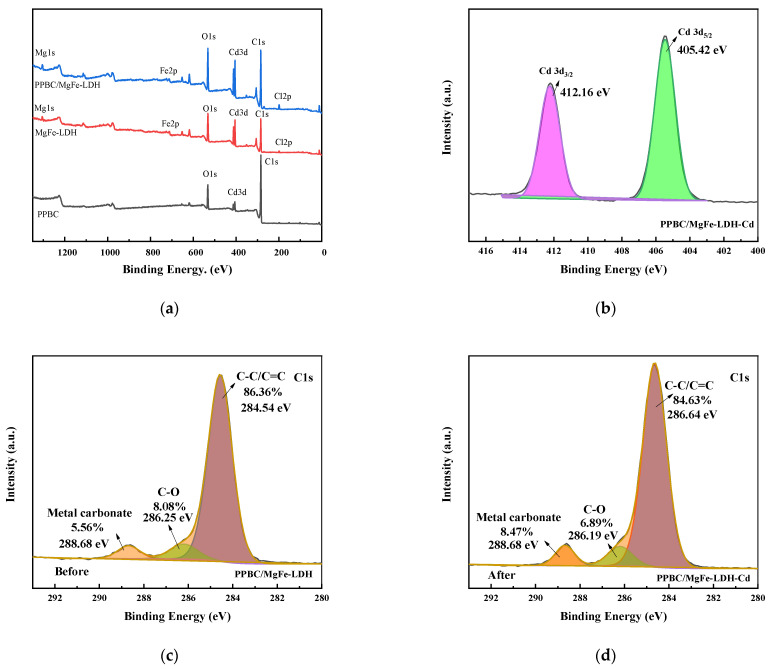
(**a**) XPS survey; (**b**) Cd3d; (**c**) C1s (before); (**d**) C1s (After); (**e**) O1s (before); (**f**) O1s (after); (**g**) Fe2p (before); (**h**) Fe2p (after) spectra of the PPBC/MgFe-LDH adsorption of Cd (II).

**Figure 13 molecules-28-04538-f013:**
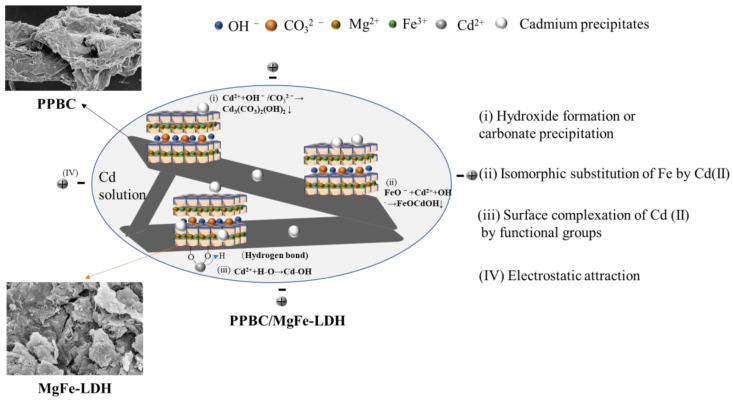
The Cd (II) adsorption mechanism of PPBC/MgFe-LDH.

**Figure 14 molecules-28-04538-f014:**
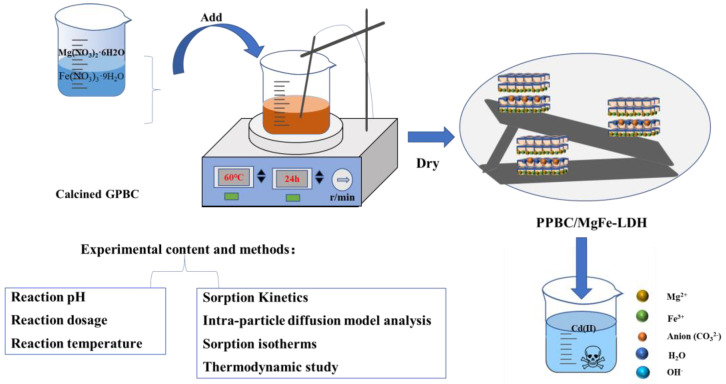
Material preparation and experimental design drawing.

**Table 1 molecules-28-04538-t001:** BET characterization of samples.

Materials	S_BET_/(m^2^·g^−1^)	V_total_/(cm^3^·g^−1^)	D_BET_/nm
PPBC	230.787	0.139	2.564
MgFe-LDH	112.716	0.109	3.619
PPBC/MgFe-LDH	20.995	0.062	12.161

**Table 2 molecules-28-04538-t002:** Elemental analysis.

Materials	C (%)	H (%)	O (%)	N (%)	S (%)
PPBC	66.960	0.920	30.180	1.690	0.250
MgFe-LDH	0.740	3.30	31.440	1.650	0.290
PPBC/MgFe-LDH	25.210	2.390	71.470	0.420	0.510

**Table 3 molecules-28-04538-t003:** Coefficients of pseudo-first-order, pseudo-second-order, and intra-particle diffusion model for the sorption of Cd (II) on the PPBC, MgFe-LDH, and PPBC/MgFe-LDH.

Adsorbent	1st Order Kinetics			2nd Order Kinetics			Experimental Value of Q	Intra-Particle Diffusion Model					
	Q_e_ (mg·g^−1^)	K_1_ (min^−1^)	R^2^	Q_e_ (mg·g^−1^)	K_2_ (g (mg·min) ^−1^)	R^2^	Q (mg·g^−1^)	K_3_ (mg·g^−1^ ·min^0.5^)	R^2^	K3′ (mg·g^−1^ ·min^0.5^)	R^2^	K3″ (mg·g^−1^ ·min^0.5^)	R^2^
PPBC	89.025	2.596 × 10^−2^	0.751	90.808	6.885 × 10^−4^	0.898	92.233	4.282	0.991	1.033	0.703	/	/
MgFe-LDH	94.847	4.390 × 10^−3^	0.799	100.838	7.462 × 10^−5^	0.872	110.620	1.050	0.984	0.286	0.970	/	/
PPBC/MgFe-LDH	182.720	2.360 × 10^−3^	0.984	223.492	9.733 × 10^−6^	0.989	189.887	0.871	0.997	3.275	0.998	0.043	−0.749

**Table 4 molecules-28-04538-t004:** Fitting parameters of adsorption isotherm.

Materials	Temperatures (°C)	Langmuir Model			Freundlich Model		
		Q_m_ (mg·g^−1^)	K_L_ (L·mg^−1^)	R^2^	K_F_ (mg·g^−1^) (mg·L^−1^)^–1/n^	1/n	R^2^
BC	25	48.498	71.492	0.851	49.912	0.094	0.806
	35	68.301	30.645	0.883	61.408	0.138	0.880
	45	91.214	11.192	0.964	65.984	0.119	0.865
MgFe-LDH	25	111.424	0.007	0.975	8.966	0.369	0.951
	35	246.171	0.103	0.940	57.995	0.271	0.907
	45	349.732	0.111	0.942	68.309	0.296	0.715
PPBC/MgFe-LDH	25	222.341	0.047	0.969	32.709	0.305	0.967
	35	226.746	0.299	0.980	43.060	0.344	0.939
	45	448.960	0.065	0.993	50.705	0.418	0.898

**Table 5 molecules-28-04538-t005:** Comparison of the maximum adsorption capacity of Cd (II) on LDHs with other adsorbents.

Adsorbents	pH	Adsorption Capacity (mg·g^−1^)	References
CS/MgAl-LDH	6	140.800	[21]
Fe-PPBC@LDH	4.5	320.650	[43]
HA/MgAl-LDH	5	155.280	[44]
Fe_3_O_4_@NiAl-LDH@guargum bionanocomposites (GLF-BNCs)	10	258	[45]
LDH/MOF	5	415.300	[46]
PPBC/MgFe-LDH	5.5	448.960	This study

**Table 6 molecules-28-04538-t006:** Thermodynamic parameters.

Materials	Temperature (K)	ΔG^0^ (kJ·mol^−1^)	ΔH^0^ (kJ·mol^−1^)	ΔS^0^[J·(mol·K)^−1^]
BC	298	−44.807	−94.164	−72.955
	308	−44.142		
	318	−42.912		
MgFe-LDH	298	−21.936	109.944	445.930
	308	−29.557		
	318	−30.714		
PPBC/MgFe-LDH	298	−26.653	14.207	141.629
	308	−32.286		
	318	−29.300		

## Data Availability

Data will be made available on request.

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
