# Peer review of "Efficient Adsorption Capacity of MgFe-Layered Double Hydroxide Loaded on Pomelo Peel Biochar for Cd (II) from Aqueous Solutions: Adsorption Behaviour and Mechanism"

_molecules, 2023, doi:10.3390/molecules28114538_

Round 1
Reviewer 1 Report
The article entitled “Efficient adsorption capacity of MgFe layered double hydroxide loaded on Pomelo peel biochar for Cd(Ⅱ ) from aqueous solutions: Adsorption behaviour and mechanism” submitted by Huang et al., describes the synthesis of biochar/MgFe-layered double hydroxide composite, its characterization, and application as adsorbent for the removal of Cd (II) ions from water. The adsorption mechanism was studied by X-ray diffraction (XRD), Fourier transforms infrared spectroscopy (FTIR), Brunauer-Emmett-Teller (BET) analysis, zeta potential analyser (Zeta), scanning electron microscopy-energy dispersive spectroscopy (SEM-EDS), and X-ray Photoelectron Spectroscopy (XPS) analyses. The article presents value-added information for the scientific community working on the development of materials for water decontamination. Most of the results have been discussed with scientific logic. Keeping in view the quantity and quality of work and its presentation, I will recommend this work to accept for publication after a successful revision as suggested below.
1. The abstract contains adsorption capacity calculated from Langmuir Adsorption model. The actual experimental adsorption capacity should also be given in the abstract.
2. The novelty of this work should be described in the introduction.
3. The discussion on Figure needs some improvements. In Figure 2(a) The wavy like support and particles on that support should be discussed. In Figure 2 (b) and 2 (c), the claimed stacking should be marked in the SEM images and origin of this stacking should be discussed.
4. In the discussion of Figure 2, it is stated that “Furthermore, partial surface pores appeared that were filled with nanosized particles.” It is clear that in which sample these partial surface pores appear and how they were visualized. Which nanosized particles were formed?
5. The possible reason for the increase in pore size due to loading of MgFe-LDH should be added in in the discussion of Table 1.
6. The term pHzpc should be stated with full name at its first appearance in section 2.2.1.
7. The statement “The Cd (II) precipitated when the pH exceeded 7” should be supported with reference. i.e., RSC Adv., 2015, 5, 43873
8. Figure 3, the negative zeta potential for MgFe-LDH increases with the increase in pH up to 6 but the adsorption efficiency is not affected in this range. This should be explained.
9. In section 2.2., the term removal rate is not suitable. It should be percentage removal. The removal rate is calculated from the kinetics study in terms of amount adsorbed as a function of time.
10. The mathematical model used for Figure should be provided or suitable reference should be given. The given mathematical models 4 and 5 have different axis titles than those used in Figure 5.
11. The experimental value of Qe should be stated in table 3 for comparison.
12. In Figure 6, three stages should be marked on the graphical line for PPBC/MgFe-LDH composite.
the overall quality of English language is good. The use subscript and superscript while writing the parameters of equation in the text need corrections.
Author Response
Dear Editors and Reviewers:
Thank you for your letter and the reviewers’ comments concerning our paper entitled” Efficient adsorption capacity of MgFe layered double hydroxide loaded on Pomelo peel biochar for Cd (Ⅱ) from aqueous solutions: Adsorption behaviour and mechanism” (Manuscript ID: molecules-2415962). Those comments are all valuable and very helpful for revising and improving our paper. According to the comments, we have revised our manuscript carefully. Revised portions are marked in red on the paper. The main correction in the paper that responds to the reviewers’ comments are as follows:
Response to Reviewer 1 Comments
Point 1: The abstract contains adsorption capacity calculated from Langmuir Adsorption model. The actual experimental adsorption capacity should also be given in the abstract.
Response 1: Thank you for your advice, I have added the actual experimental adsorption capacity in the abstract.
Point 2: The novelty of this work should be described in the introduction.
Response 2: Thank you for your advice: I have described novelty of this work in the introduction: Many researchers reported biochar and LDH materials for mental removal. Yang Jia[31] synthesized magnetic biochar supporting MgFe-LDH composites remove Pb2+ from the aqueous solution, the adsorption process was aspontaneous endothermic reaction and limited by chemisorption. Tao Wang[32] concentrated on loading of MnAl-LDH on biochar for Cu (ΙΙ) treatment, biochar exhibited higher adsorption capacity than most modified biochar and activated carbon. Compared with these materials, the biochar/LDH shows multiple advantages such as simple synthesis, good adsorption performance and easy separation. Besides, up to now, the quantitative analysis of heavy metal removal mechanisms is mainly focused on LDH or BC materials and there is little research on other materials, such as BC-LDH composite. Therefore, the combination of biocarbon materials with LDH in a composite material with a novel structure and advanced surface properties is of practical use for engineering purposes.
Point 3: The discussion on Figure needs some improvements. In Figure 2(a) The wavy like support and particles on that support should be discussed. In Figure 2 (b) and 2 (c), the claimed stacking should be marked in the SEM images and origin of this stacking should be discussed.
Response 3: Thank you for your advice, we added discussion about figure 2(a): the wavy like support and particles on that support is the skeleton of the grapefruit skin biochar expands into fractures, the structure is similar to Wang’s [29] result. In Figure 2 (b) and 2 (c), the claimed stacking has been marked in the SEM images.
Point 4: In the discussion of Figure 2, it is stated that “Furthermore, partial surface pores appeared that were filled with nanosized particles.” It is clear that in which sample these partial surface pores appear and how they were visualized. Which nanosized particles were formed?
Response 4: Thank you for your advice, The EDS spectra in Figure 3(d) can be used to judge the presence of nanoparticles in the surface pores of PPBC/MgFe-LDH. In the surface and pores of PPBC/MgFe-LDH material, the orange highlight represents C element, the green highlight represents Mg element, and the purple highlight represents iron element. Enlarge figure 3(d), it can be observed that the green and purple bright spots in the pores of the structure are evenly distributed, so it can be determined that MgFe-LDH nanoparticles are dispersed in the surface pores.
Point 5: The possible reason for the increase in pore size due to loading of MgFe-LDH should be added in in the discussion of Table 1.
Response 5: Thank you for your advice, i have added the reasonable discussion of aperture enlargement: The reason for the increased pore size of MgFe-LDH after loading can be combined with the SEM-EDS results, MgFe-LDH being loaded on the surface and within the pores of the PPBC, thereby reducing the total pore area and pore volume and causing the collapse of the structure after loading such that new and larger pores are created. This result may differ from Yang Jia’s synthesis path [31], but the pore size of Wang's[29] grapefruit skin biochar also increased after loading loads nano-γ-Fe2O3 particles.
Point 6: The term pHzpc should be stated with full name at its first appearance in section 2.2.1.
Response 6: Thank you for your advice, I have stated pHzpc with full name pH-Zero point of charge(pHzpc).
Point 7: The statement “The Cd (II) precipitated when the pH exceeded 7” should be supported with reference. i.e., RSC Adv., 2015, 5, 43873
Response 7: Thank you for your advice, I have added the supported reference you recommended.
Point 8: Figure 3, the negative zeta potential for MgFe-LDH increases with the increase in pH up to 6 but the adsorption efficiency is not affected in this range. This should be explained.
Response 8: Thank you for your advice, In the range of 4-6, the MgFe-LDH demonstrated a stable removal efficiency, may be that electrostatic adsorption is not the main way of adsorption for LDH, and precipitation is the main way. Thus, the change of removal rate is small.
Point 9: In section 2.2., the term removal rate is not suitable. It should be percentage removal. The removal rate is calculated from the kinetics study in terms of amount adsorbed as a function of time.
Response 9: Thank you for your advice, I have repalced removal rate with adsorption capacity.
Point 10: The mathematical model used for Figure should be provided or suitable reference should be given. The given mathematical models 4 and 5 have different axis titles than those used in Figure 5.
Response 10: Thank you for your advice, I have unified the axis titles for models 4 and 5. And the suitable reference is added.
Point 11: The experimental value of Qe should be stated in table 3 for comparison.
Response 11: Thank you for your advice, I have stated experimental value of Qe in table 3 for comparison.
Point 12: In Figure 6, three stages should be marked on the graphical line for PPBC/MgFe-LDH composite.
Response 12: Thank you for your advice, I have marked three stages in the figure 6.

Reviewer 2 Report
This manuscript presents quite of of interesting findings, yet some modifications and clarifications should be done. I would recommend publishing the manuscript after some modifications.
The added comments are not by any means to diminish the authors’ job, yet to improve the quality of the manuscript.
Comments:
L31: mercury instead of hhdrargyrum
L31: remove non-metallic
L32-34: combine the two phrases
L43-L44: add a refernce to support your statement
L65: replace “compounding” by “functionalising”
Suggested ref:
https://www.mdpi.com/2310-2861/9/4/327
https://www.mdpi.com/2310-2861/9/4/304
L73-L74: support the statement with a reference
L75-L77: same
L90: remove extensively
Results:
-The authors claimed that 002 is representative of cellulose, even know the signal is more likely is in the base noise region. Cellulose presents an ubiquitous character along plants. Following FTIR analysis no broad range stretching has been yielded 1000 cm-1. How do you explain that?
L114-115: it could also be from the hydroxyl function in cellulose
- I would really doubt that OH is showing a bond stretching at 1626 cm-1. This scissored peak could relate to a carbonyl function of an amide. Or the deconvolution could be simply due to the lower sensibility of the device.
- The carbonate broad range is more likely on 1400 cm-1
- Specify if the bias is being obtained from the instruments’ sensibility.
- L149: Why to add “Finally”?
- L149-L154: how the bulk analysis confirm the targeted synthesis. Support your statement with discussion to already published findings.
- L161: increasingly likely is not correct. Modify.
- L162: the role of the pH is not absolute
- L172-L174: this analysis is for the blue curve. If yes, it is not PPBC.
- L179: when the pH is greater than 4 not 4.03. If this not an interpretation of Fig 3b, you should add a ref so support your statement.
- L188-L199: no need to mention the data point by point, the increase is obvious. I did not get what are you trying to prove; indeed the efficiency will increase when the dose is being increased. You should more likely highlight the efficiency of the PPBC/MgFe-LDH as a higher efficiency is being yielded at lower doses if compared with two other materials. On the other hand, at 1.5 g/L the investigated material would hit a plateau, indicating the saturation of the active sites.
- Figure 5: the full and dashed line presents kinetics for PPBC only or for the other materials as well? If it is the second case, this point should clarified in the caption.(same for all similar figures)
- Table 3: some numbers are not clear, if it is not the pdf error of conversion, recheck.
- L260: chemisoprtion health ?
- Figure 8b; same comments as above
M&M:
- A part for the detailed characterization methods should be added (pursuing company, conditions of the device, conditioning of the sample…).
- 4.4: organise the equation parts in a more suited way. It is worth it to split this part into 3 one for the kinetic model, langumir adsorption, freundlich adsorption. Add citation to the different parts.
Could be approved for abstract and introduction, yet acceptable.
Author Response
Dear Editors and Reviewers:
Thank you for your letter and the reviewers’ comments concerning our paper entitled” Efficient adsorption capacity of MgFe layered double hydroxide loaded on Pomelo peel biochar for Cd (Ⅱ) from aqueous solutions: Adsorption behaviour and mechanism” (Manuscript ID: molecules-2415962). Those comments are all valuable and very helpful for revising and improving our paper. According to the comments, we have revised our manuscript carefully. Revised portions are marked in red on the paper. The main correction in the paper that responds to the reviewers’ comments are as follows:
Point 1: L31: mercury instead of hydrargyrum
Response 1: Thank you for your advice, I have replaced the mercury with hydrargyrum.
Point 2: L31: remove non-metallic
Response 3: Thank you for your advice, I have removed non-metallic
Point 3: L32-34: combine the two phrases
Response 3: Thank you for your advice, I have combined the two phrases: They are non-biodegradable contaminants that can accumulate in the food chain and cause sickness or contribute to chronic disorders.
Point 4: L43-L44: add a refernce to support your statement
Response 4: Thank you for your advice, i have added references to support our statement: Several high-efficiency adsorbents are limited in application considering their high prices: such as activated carbon is expensive and requires chelating agents to enhance the performance of metal sorption, thus increasing the cost of treatment [7].
Point 5: L65: replace “compounding” by “functionalising”
Response 5: Thank you for your advice, I have replaced compounding with functionalizing.
Point 6: Suggested ref:
https://www.mdpi.com/2310-2861/9/4/327
https://www.mdpi.com/2310-2861/9/4/304
Response 6: Thank you for your advice, I have cited the suggested reference.
Point 7: L73-L74: support the statement with a reference
Response 7: Thank you for your advice, I have added the support reference: However, considering the high cost of their preparation, several highly efficient carbon materials have limited applications. Such as multi- and single-walled car-bon nanotubes (CNTs), carbon quantum dots, graphene, and date palm ash, have been studied previously to extend their properties and functionalities [25-28].
Point 8: L75-L77: same
Response 8: Thank you for your advice, I have added the support reference: It is a kind of porous biomass rich in cellulose and pectin [29]. Although the annual output of pomelo peel is huge, but only a small amount of pomelo peel is reprocessed into medicine or used as chemical raw material. Most pomelo peel is treated as agricultural waste, which not only causes great waste of resources, but also causes pollution to the environment. According to statistics, the annual output of pomelo in China is more than 4.5 million tons, and pomelo peel accounts for 30% - 40% of the total quality of pomelo, so the amount of waste pomelo peel produced every year is 1.35 million ~ 1.8 million tons [30].
Point 9: L90: remove extensively
Response 9: Thank you for your advice, I have removed extensively.
Results:
Point 1: The authors claimed that 002 is representative of cellulose, even know the signal is more likely is in the base noise region. Cellulose presents an ubiquitous character along plants. Following FTIR analysis no broad range stretching has been yielded 1000 cm-1. How do you explain that?
Response 1: Thank you for your advice, PPBC displayed a broad and strong diffraction peak around 23°, matching the (002) crystal plane of cellulose, which is typical of amorphous carbon components [31], indicating that there was an obvious aromatic carbon structure on the surface[29]. According to FTIR ,PPBC have characteristic peaks of aromatic group -CH2- (1387 cm-1), C-O (1096 cm-1)[29].
Point 2: L114-115: it could also be from the hydroxyl function in cellulose. I would really doubt that OH is showing a bond stretching at 1626 cm-1. This scissored peak could relate to a carbonyl function of an amide. Or the deconvolution could be simply due to the lower sensibility of the device.The carbonate broad range is more likely on 1400 cm-1. Specify if the bias is being obtained from the instruments’ sensibility.
Response 2: Thank you for your advice, a bond stretching at 1626 cm-1 was due to the bending vibration of H–O–H(δH − O − H), and it should be assigned to the adsorbed water molecule in the interlayer [36]. Compared with that in PPBC, the existence of interlayer carbonate can be demonstrated via the asymmetric stretching absorption band (ν3) of the CO32− close to 1380 cm−1 and 1400 cm−1, suggesting the successful modification of MgFe-LDH on PPBC [37].
Point 3: L149: Why to add “Finally”?
Response 3: Thank you for your advice, adding “Finally” is because elemental analysis is the last characterization we made, and the word has been removed here.
Point 4: L149-L154: how the bulk analysis confirm the targeted synthesis. Support your statement with discussion to already published findings.
Response 4: Thank you for your advice, for the MgFe-LDH composite with PPBC, the oxygen content increased from 31.440 % to 71.470 %, and the carbon content increased from 0.740 % to 25.210 %. This result is similar to Zhou’s study[29].
Point 5: L161: increasingly likely is not correct. Modify.
Response 5: Thank you for your advice, i have corrected the inappropriate expression:
Point 6: L162: the role of the pH is not absolute
Response 6: Thank you for your advice, I have changed the absolute role to important role.
Point 7: L172-L174: this analysis is for the blue curve. If yes, it is not PPBC.
Response 7: Thank you for your advice, the analysis is for PPBC with dark curve.
Point 8: L179: when the pH is greater than 4 not 4.03. If this not an interpretation of Fig 3b, you should add a ref to support your statement.
Response 8: Thank you for your advice, I have added a ref to support my statement.
Point 9: L188-L199: no need to mention the data point by point, the increase is obvious. I did not get what are you trying to prove; indeed the efficiency will increase when the dose is being increased. You should more likely highlight the efficiency of the PPBC/MgFe-LDH as a higher efficiency is being yielded at lower doses if compared with two other materials. On the other hand, at 1.5 g/L the investigated material would hit a plateau, indicating the saturation of the active sites.
Response 9: Thank you for your advice, i have made a new explanation for the effect of dosage as follows: Compared with PPBC and MgFe-LDH, PPBC/MgFe-LDH has higher removal efficiency at low dose. At 0.5 g·L-1, the removal efficiency of PPBC/MgFe-LDH (29.611%) is four times and five times of that of PPBC (5.211%) and MgFe-LDH (7.433%). At 1.5 g·L-1, PPBC/MgFe-LDH under study reaches a plateau at a concentration of 300 mg·L-1, so we will examine the influence of temperature later in the experiment.
Point 10: Figure 5: the full and dashed line presents kinetics for PPBC only or for the other materials as well? If it is the second case, this point should be clarified in the caption. (Same for all similar figures)
Response 10: Thank you for your advice, I have clarified the meaning of full and solid line presents in the caption.
Point 11: Table 3: some numbers are not clear, if it is not the pdf error of conversion, recheck.
Response 11: Thank you for your advice, numbers have been checked and is clear now.
Point 12: L260: chemisoprtion health?
Response 12: Thank you for your advice, i have corrected my incorrect expression it should be chemisorption bonds.
Point 13: Figure 8b; same comments as above
Response 13: Thank you for your advice, as for Figure 8 (b), I have modified and added the above questions.
M&M:
Point 1: A part for the detailed characterization methods should be added (pursuing company, conditions of the device, conditioning of the sample…).
Response 1: Thank you for your advice, I have added more detailed characterization methods including instrument company, equipment condition and sample condition.
Point 2: 4.4: organise the equation parts in a more suited way. It is worth it to split this part into 3 one for the kinetic model, langumir adsorption, freundlich adsorption. Add citation to the different parts.
Response 2: Thank you for your advice, I have reorganized the equation parts into 2 parts: kinetic model and Isotherm model, and added citation to the different parts.

Reviewer 3 Report
In this interesting study, the authors report the preparation and characterization of a novel adsorbent for removing cadmium ions. Isotherm, kinetic and mechanism studies for explaining the adsorption process were conducted. This study is highly relevant as content and logically built. I recommend that some points are addressed:
- Introduction: The authors should better explain why pomelo was used as an agroindustrial waste to prepare adsorbent. Moreover, I recommend add technical information and actual references for such purpose.
- Introduction: How is this system different to other reports to merit publication? Please, report.
- Results and discussion: The authors should represent all experimental results as mean ± error deviation.
- Results and discussion: Desorption and regeneration studies (influence of several desorption reactants; influence of desorption agent concentration and reusability tests) shoul be incorporated to this study. It is very interesting to the readers.
In this interesting study, the authors report the preparation and characterization of a novel adsorbent for removing cadmium ions. Isotherm, kinetic and mechanism studies for explaining the adsorption process were conducted. This study is highly relevant as content and logically built. I recommend that some points are addressed:
- Introduction: The authors should better explain why pomelo was used as an agroindustrial waste to prepare adsorbent. Moreover, I recommend add technical information and actual references for such purpose.
- Introduction: How is this system different to other reports to merit publication? Please, report.
- Results and discussion: The authors should represent all experimental results as mean ± error deviation.
- Results and discussion: Desorption and regeneration studies (influence of several desorption reactants; influence of desorption agent concentration and reusability tests) shoul be incorporated to this study. It is very interesting to the readers.
Author Response
Dear Editors and Reviewers:
Thank you for your letter and the reviewers’ comments concerning our paper entitled” Efficient adsorption capacity of MgFe layered double hydroxide loaded on Pomelo peel biochar for Cd (Ⅱ) from aqueous solutions: Adsorption behaviour and mechanism” (Manuscript ID: molecules-2415962). Those comments are all valuable and very helpful for revising and improving our paper. According to the comments, we have revised our manuscript carefully. Revised portions are marked in red on the paper. The main correction in the paper that responds to the reviewers’ comments are as follows:
Point 1: Introduction: The authors should better explain why pomelo was used as an agroindustrial waste to prepare adsorbent. Moreover, I recommend add technical information and actual references for such purpose.
Response 1: Thank you for your advice, I have added the information : pomelo peel biochar is a kind of porous biomass rich in cellulose and pectin. The annual output of pomelo peel is huge, but only a small amount of pomelo peel is reprocessed into medicine or used as chemical raw material. Most pomelo peel is treated as agricultural waste, which not only causes great waste of resources, but also causes pollution to the environment. According to statistics, the annual output of pomelo in China is more than 4.5 million tons, and pomelo peel accounts for 30% ~ 40% of the total quality of pomelo, so the amount of waste pomelo peel produced every year is 1.35 million ~ 1.8 million tons.
Point 2: Introduction: How is this system different to other reports to merit publication? Please, report.
Response 2: Thank you for your advice: I have described novelty of this work in the introduction: Many researchers reported biochar and LDH materials for mental removal. Yang Jia[31] synthesized magnetic biochar supporting MgFe- LDH composites remove Pb2+ from the aqueous solution, the adsorption process was aspontaneous endothermic re-action and limited by chemisorption. Tao Wang[32] concentrated on loading of MnAl-LDH on biochar for Cu (ΙΙ) treatment, biochar exhibited higher adsorption ca-pacity than most modified biochar and activated carbon. Compared with these mate-rials, the biochar/LDH shows multiple advantages such as simple synthesis, good ad-sorption performance and easy separation. Besides, up to now, the quantitative analy-sis of heavy metal removal mechanisms is mainly focused on LDH or BC materials and there is little research on other materials, such as BC-LDH composite. Therefore, the combination of biocarbon materials with LDH in a composite material with a novel structure and advanced surface properties is of practical use for engineering purposes.
Point 3: Results and discussion: The authors should represent all experimental results as mean ± error deviation.
Response 3: Thank you for your advice, I have expressed all experimental results as mean ± error deviation.
Point 4: Results and discussion: Desorption and regeneration studies (influence of several desorption reactants; influence of desorption agent concentration and reusability tests) should be incorporated to this study. It is very interesting to the readers.
Response 4: Thank you very much for pointing out this important issue. We agree with your comments that preliminary experiments are also necessary. Your suggestion provides a direction for our next research. Unfortunately, due to the limited time and funding, we did not supplement experimental validation. The following research will focus on this aspect.

Reviewer 4 Report
Manuscript Title: Efficient adsorption capacity of MgFe layered double hydrox-ide loaded on Pomelo peel biochar for Cd(Ⅱ) from aqueous so-lutions: Adsorption behaviour and mechanism
In the present work, The prepared pomelo peel biochar/MgFe-layered double hydroxide composite through co-precipitation to remove Cd (II) from aqueous solutions was investigated. The authors also investigated the chemical and physical characteristics of pomelo peel biochar/MgFe-layered double hydroxide composite.
The writing is well tailored, and the explanations are often reasonable. However, sometimes, at some points, the submitted research leaves something to be desired Nevertheless, the presented manuscript could be recommended for the publication after a major revision and amendments following these next comments:
Comment #1: you should write the first occurrence of phrases in full, then you can use the abbreviations or acronyms throughout the rest of the manuscript such as PPBC.
Comment #2: In abstract, author should avoid the use of very long sentences and abbreviations phrases in full, and abstract should be supported with more quantitative results.
Comment #3: Introduction section should be supported with more up-to-date references, the literature review needs more improvement by describing the roles of MgFe on the contaminant’s removal as well as the Cd (II) removal mechanism, the following papers should be cited to support that dissection, (Insights into boron removal from water using Mg-Al-LDH: Reaction parameters optimization & 3D-RSM modeling, Journal of Water Process Engineering 46, 102608).
Comment #4: The main objectives and novelty aspects in this study should be highlighted clearly within the introduction section.
Comment #5: Authors should estimate the degree of crystallinity (DOC%) and crystallite size of the developed nanocomposite based on the obtained XRD patterns, as they are important features which could affect the reactive performance and the removal mechanisms. Hence, the following references should be considered: (Multi-functional magnesium hydroxide coating for iron nanoparticles towards prolonged reactivity in Cr (VI) removal from aqueous solutions, Journal of Environmental Chemical Engineering 10 (3), 107431)
Comment #6: Coefficient of determination, "R squared" is low accuracy comparison factor, the author has to use other mothed such as Akaike’s Information Criterion (AIC) which a commonly used as statistical approach to compare between different models. In this regard, the following reference could be useful: (A novel method to improve methane generation from waste sludge using iron nanoparticles coated with magnesium hydroxide, Renewable and Sustainable Energy Reviews 158, 112192)
Comment #7: Thermodynamics analysis is required to support the assumption of removal mechanism. In this regard, the following reference could be useful: (Insights into boron removal from water using Mg-Al-LDH: Reaction parameters optimization & 3D-RSM modeling, Journal of Water Process Engineering 46, 102608) and (Multi-functional magnesium hydroxide coating for iron nanoparticles towards prolonged reactivity in Cr (VI) removal from aqueous solutions, Journal of Environmental Chemical Engineering 10 (3), 107431)
Comment #8: Kinetics, adsorption and isotherm analysis is required to support the assumption of removal mechanism. In this regard, the following reference could be useful: (Multi-functional magnesium hydroxide coating for iron nanoparticles towards prolonged reactivity in Cr (VI) removal from aqueous solutions, Journal of Environmental Chemical Engineering 10 (3), 107431)
Comment #9: The chemical pathways involved in the removal mechanisms should be summarized and discussed in detail (supported with illustrating reaction equations, if possible).
Comment #10: Methodology need more clarification and explanation a detailed schematic of reactors design should be included in the methodology section.
Comment #11: Did the authors duplicate all experiments? If yes, error bars should be provided in all charts.
Comment #12: Reusability and recycling experiments should be conducted to the reacted composite.
Comment #13: The effect of reaction temperature should be investigated as well.
Comment #14: Removal of Cd(Ⅱ) from real contaminated water by the proposed system should be investigated to sure its real scale applications.
Comment #15: The author needs to investigate the effect of co-existing anions and co-existing cations
Comment #16: Conclusions should be supported with more qualitative findings of the study.
Comment #17: References should be revised to ensure that volume, and start-ending pages are provided, whenever possible.
Comment #18: The authors should consider revising the whole text formatting in the manuscript for any additional spacing, words capitalization, unifying the font, and unnecessary repetitions.
Comment #19: I would like to see the manuscript again after the revision
Comment #1: The used font type should be unified within the whole manuscript (starting with the title).
Comment #2: The use of English grammar still requires some work on the whole manuscript.
Author Response
Dear Editors and Reviewers:
Thank you for your letter and the reviewers’ comments concerning our paper entitled” Efficient adsorption capacity of MgFe layered double hydroxide loaded on Pomelo peel biochar for Cd (Ⅱ) from aqueous solutions: Adsorption behaviour and mechanism” (Manuscript ID: molecules-2415962). Those comments are all valuable and very helpful for revising and improving our paper. According to the comments, we have revised our manuscript carefully. Revised portions are marked in red on the paper. The main correction in the paper that responds to the reviewers’ comments are as follows:
Comment #1: you should write the first occurrence of phrases in full, then you can use the abbreviations or acronyms throughout the rest of the manuscript such as PPBC.
Response 1: Thank you for your advice, I've checked and modified phrases and acronyms that first appear.
Comment #2: In abstract, author should avoid the use of very long sentences and abbreviations phrases in full, and abstract should be supported with more quantitative results.
Response 2: Thank you for your advice, i have modified the sentence format of the abstract and added some quantitative results to support it.
Comment #3: Introduction section should be supported with more up-to-date references, the literature review needs more improvement by describing the roles of MgFe on the contaminant’s removal as well as the Cd (II) removal mechanism, the following papers should be cited to support that dissection, (Insights into boron removal from water using Mg-Al-LDH: Reaction parameters optimization & 3D-RSM modeling, Journal of Water Process Engineering 46, 102608).
Response 3: Thank you for your advice, I've cited the recommended references and describes the role of MgFe-LDH in removing Cd (II) and the shortcomings of the removal mechanism: However, the locking of Cd (II) adsorption on LDH relies primarily on precipitation and is strongly influenced by the environment and pH of the material surface, result-ing in poor adsorption stability [20]. The LDH surface is usually positively charged, which reduces the adsorption locking of Cd (II) by electrostatic repulsion, thereby se-verely limiting the adsorption of heavy metal cations [21].The adsorption performance may be further improved by functionalising LDH with other materials such that their surface physicochemical properties may be modified [21, 22].
Comment #4: The main objectives and novelty aspects in this study should be highlighted clearly within the introduction section.
Response 4: Thank you for your advice: I have described novelty of this work in the introduction: Many researchers reported biochar and LDH materials for mental removal. Yang Jia[29] synthesized magnetic biochar supporting MgFe- LDH composites remove Pb2+ from the aqueous solution, the adsorption process was aspontaneous endothermic reaction and limited by chemisorption. Tao Wang[30] concentrated on loading of MnAl-LDH on biochar for Cu (ΙΙ) treatment, biochar exhibited higher adsorption capacity than most modified biochar and activated carbon. Compared with these materials, the biochar/LDH shows multiple advantages such as simple synthesis, good adsorption performance and easy separation. Besides, up to now, the quantitative analysis of heavy metal removal mechanisms is mainly focused on LDH or BC materials and there is little research on other materials, such as BC-LDH composite. Therefore, the combination of biocarbon materials with LDH in a composite material with a novel structure and advanced surface properties is of practical use for engineering purposes.
Response 4:
Comment #5: Authors should estimate the degree of crystallinity (DOC%) and crystallite size of the developed nanocomposite based on the obtained XRD patterns, as they are important features which could affect the reactive performance and the removal mechanisms. Hence, the following references should be considered: (Multi-functional magnesium hydroxide coating for iron nanoparticles towards prolonged reactivity in Cr (VI) removal from aqueous solutions, Journal of Environmental Chemical Engineering 10 (3), 107431)
Response 5: Thank you for your advice: According to the XRD data, we can calculate the crystallinity and grain size of the prepared nanocomposites. Crystallinity is an important feature affecting reaction performance and removal mechanism [32]. The crystallinity calculated by software is relative crystallinity. Calculated by jade software, the crystallinity of PPBC/MgFe-LDH is 28.13%, crystallite size is 124Å.
Comment #6: Coefficient of determination, "R squared" is low accuracy comparison factor, the author has to use other mothed such as Akaike’s Information Criterion (AIC) which a commonly used as statistical approach to compare between different models. In this regard, the following reference could be useful: (A novel method to improve methane generation from waste sludge using iron nanoparticles coated with magnesium hydroxide, Renewable and Sustainable Energy Reviews 158, 112192)
Response 6: Thank you for your advice, the methods we selected are all classical methods, which are basically used for adsorption. The new simulation method of this paper is indeed quite good, but we are relatively unfamiliar with the model at present. In the following research, we will try to use the modified method to compare the models.
Comment #7: Thermodynamics analysis is required to support the assumption of removal mechanism. In this regard, the following reference could be useful: (Insights into boron removal from water using Mg-Al-LDH: Reaction parameters optimization & 3D-RSM modeling, Journal of Water Process Engineering 46, 102608) and (Multi-functional magnesium hydroxide coating for iron nanoparticles towards prolonged reactivity in Cr (VI) removal from aqueous solutions, Journal of Environmental Chemical Engineering 10 (3), 107431)
Response 7: Thank you for your advice: I've added thermodynamic analysis and cited the reference you recommended.
Comment #8: Kinetics, adsorption and isotherm analysis is required to support the assumption of removal mechanism. In this regard, the following reference could be useful: (Multi-functional magnesium hydroxide coating for iron nanoparticles towards prolonged reactivity in Cr (VI) removal from aqueous solutions, Journal of Environmental Chemical Engineering 10 (3), 107431)
Response 8: Thank you for your advice and recommendation,kinetic and isothermal analyses were performed to support the hypothesis of removal mechanism and cited the reference you recommended.
Comment #9: The chemical pathways involved in the removal mechanisms should be summarized and discussed in detail (supported with illustrating reaction equations, if possible).
Response 9: Thank you for your advice, I added the reaction equation involving chemical pathways to the mechanism section.
Comment #10: Methodology need more clarification and explanation a detailed schematic of reactors design should be included in the methodology section.
Response 10: Thank you for your advice, we have supplemented the concrete characterization methods of materials, and supplemented the detailed schematic diagram of the material preparation process in the methodology section.
Comment #11: Did the authors duplicate all experiments? If yes, error bars should be provided in all charts.
Response 11: Thank you for your advice, all the experiments were duplicated, and error bars were provided in all charts.
Comment #12: Reusability and recycling experiments should be conducted to the reacted composite.
Response 12: Thank you very much for pointing out this important issue. We agree with your comments that preliminary experiments are also necessary. Your suggestion provides a direction for our next research. Unfortunately, due to the limited time and funding, we did not supplement experimental validation. The following research will focus on this aspect.
Comment#13: The effect of reaction temperature should be investigated as well.
Response 13: Thank you for your advise, we have added the effect of reaction temperature on material adsorption.
Comment #14: Removal of Cd(Ⅱ) from real contaminated water by the proposed system should be investigated to sure its real scale applications.
Response 14: Thank you very much for pointing out this important issue. We agree with your comments that preliminary experiments are also necessary. At present, the concentration we choose is as close to the actual concentration as possible. Your suggestion provides the direction for our next research.
Comment #15: The author needs to investigate the effect of co-existing anions and co-existing cations
Response 15: Thank you for your advice, we are very sorry that we failed to investigate the effect of co-existing anions and co-existing cations in this study, we aims to investigate the mechanism of Cd(Ⅱ) adsorption by composite materials. Therefore, we are mainly concerned with the results of the adsorption mechanism. We added this deficiency to the restriction section, which we will investigate further in the future.
Comment #16: Conclusions should be supported with more qualitative findings of the study.
Response 16: Thank you for your advise, we have modified the conclusions:
Comment #17: References should be revised to ensure that volume, and start-ending pages are provided, whenever possible.
Response 17: Thank you for your advise, I have checked all the reference and ensure that volume, and start-ending pages are provided.
Comment #18: The authors should consider revising the whole text formatting in the manuscript for any additional spacing, words capitalization, unifying the font, and unnecessary repetitions.
Response 18: Thank you for your advise, I have checked the whole text formatting in the manuscript including spacing, words capitalization, unifying the font, and unnecessary repetitions.
Comment #19: I would like to see the manuscript again after the revision
Response 19: Thank you , I have try my best to revise my manuscript.
Comments on the Quality of English Language
Comment #1: The used font type should be unified within the whole manuscript (starting with the title). Response 1: Thank you for your advise, I have unified the font of the full text
Comment #2: The use of English grammar still requires some work on the whole manuscript.
Response 2: Thank you for your advise, I have corrected the improper English grammar in the whole text.

Round 2
Reviewer 4 Report
The authors have improved the manuscript well and acceptable in its present form
The authors have improved the manuscript well and acceptable in its present form